# Comparing Performance of NAF and NT-2013 to SGA as Nutritional Assessment Tools in Systemic Sclerosis Patients

**DOI:** 10.3390/life15081325

**Published:** 2025-08-20

**Authors:** Kittipadh Boonyavarakul, Chingching Foocharoen, Orathai Wantha, Veeradej Pisprasert

**Affiliations:** 1Department of Medicine, Faculty of Medicine, Khon Kaen University, Khon Kaen 40002, Thailand; kittipadh@gmail.com; 2Division of Rheumatology, Department of Medicine, Faculty of Medicine, Khon Kaen University, Khon Kaen 40002, Thailand; fching@kku.ac.th; 3Nursing Division, Faculty of Medicine, Khon Kaen University, Khon Kaen 40002, Thailand; orawan1@kku.ac.th; 4Division of Clinical Nutrition, Department of Medicine, Faculty of Medicine, Khon Kaen University, Khon Kaen 40002, Thailand

**Keywords:** systemic sclerosis, malnutrition, subjective global assessment (SGA), Nutritional Assessment Form (NAF), Nutritional Triage 2013 (NT-2013)

## Abstract

Malnutrition is one of the common complications of patients with systemic sclerosis (SSc). However, several nutritional assessment tools are implemented in Thailand. The study aimed to compare the performance of nutritional assessment tools including Nutritional Assessment Form (NAF) and Nutritional Triage 2013 (NT-2013) to Subjective Global Assessment (SGA) in SSc patients. A cross-sectional diagnostic study was conducted in adult SSc patients at Srinagarind Hospital, Thailand. To elucidate the efficacy and correlations of these tools, descriptive statistics, Pearson correlation analyses, and kappa coefficient of agreement were employed. A total of 208 SSc patients were included, of which 70.7% were females. The respective mean age and body mass index was 59.3 years and 21.1 kg/m^2^. Nearly half (45.7%) were malnourished based on SGA. Malnutrition diagnosis using the NAF and NT-2013 criteria were found in 80.3% and 34.6%, respectively. The respective sensitivity and specificity of NAF for diagnosis of malnutrition was 93.7% and 31.9%, while NT-2013 was 60.0% and 90.3%. Both NAF and NT-2013 had slight agreement with SGA with a kappa of 0.149 for NAF and 0.131 for NT-2013. Adjusting the cut-off points of NAF and NT-2013 could enhance sensitivity, specificity, and improve agreement for diagnosis with SGA.

## 1. Introduction

Systemic sclerosis (SSc) is a rare immune-mediated connective tissue disease with a prevalence of 17.6 per 100,000 around the world [1]. Explained by its pathophysiology, vasculopathy and diffuse fibrosis, the disease affects multiple systems of the body leading to high rates of morbidity and mortality [2,3,4]. Cardiovascular and pulmonary complications were considered the primary cause of death in systemic sclerosis patients. However, a large number of evidence-based treatments have been developed, leading to a notable reduction in mortality associated with such complications [4,5]. While many serious complications are now treatable [4], non-lethal manifestation steps up as the new challenge in improving the patient’s quality of life. One of the most common clinical features in systemic sclerosis is the gastrointestinal involvement, often caused by dysmotility and fibrosis, which affects approximately 90% of the patients with systemic sclerosis [6,7,8]. These symptoms, including dysphagia, malabsorption, constipation, diarrhea, etc., not only reduce the patients’ quality of life [9], but also put them at risk of malnutrition [4,6,9,10]. Since malnutrition was proved to be associated with morbidity and mortality, it has become essential that all systemic sclerosis patients must be screened for malnutrition [8,9,11,12].

In the past, malnutrition was assessed based on history taking, physical examination and objective parameters including anthropometric measurements, such as mid-arm circumference, and laboratory results, such as albumin level and total lymphocyte count, etc. The parameters could be interfered by various illness-related factors leading to a large number of misdiagnoses of malnutrition [13]. However, in 1987, after the construction of Subjective Global Assessment (SGA), nutritional assessment has been easier and more convenient for all medical staff [14]. Malnutrition has been assessed and diagnosed more accurately since then. While SGA has long been considered the gold standard, its subjective nature and time-consuming process have led to the exploration of alternative tools for more convenient clinical use. Over the past decades, many other new nutritional assessment tools have been announced both in Thailand and internationally. The commonly used nutritional assessment tools in Thailand include Nutritional Alert Form (NAF) and Nutritional Triage 2013 (NT-2013). Both are clinical scoring systems developed by Thai experts [13,15].

NAF is designed to be easy, concise, and does not require specialized nutrition expertise. Additionally, it can be utilized in settings where body weight measurement may not be feasible, as it incorporates the effects of serum albumin and total lymphocyte count [13]. NT-2013 consists of nine questions including dietary history, changes in body weight, fluid retention, loss of subcutaneous fat, loss of muscle mass, muscle function, chronic illness severity, acute illness severity, and a summarized score for each category [15].

As far as our concern, the nutritional assessment tools mentioned were based on experience of other diseases and there were no specific tools for assessing malnutrition in systemic sclerosis [2,6]. The aim of the study is to compare, in systemic sclerosis patients, the performances of nutritional assessment tools including NAF and NT-2013 to SGA, which is now a gold standard in diagnosing malnutrition but inconvenient in clinical practice.

## 2. Materials and Methods

A cross-sectional study enrolled systemic sclerosis patients from the scleroderma clinic, Srinagarind hospital, Thailand, between 1 May 2022 and 31 January 2024. The inclusion criteria were patients diagnosed with systemic sclerosis according to ACR/EULAR criteria 2013, aged 18 years or older and able to provide informed consent. Regarding ACR/EULAR 2013 criteria, skin thickening of the fingers extending proximal to the metacarpophalangeal joints is sufficient to be classified as SSC; if not, seven additive items are considered with different weights for each: skin thickening of the fingers, finger-tip lesions, telangiectasia, abnormal nailfold capillaries, interstitial lung disease or pulmonary arterial hypertension, Raynaud’s phenomenon, and SSc-related autoantibodies. Patients with critically ill conditions or patients who could not undergo nutritional assessment were excluded. In each patient, baseline characteristic data including age, gender, signs and symptoms of SSc as well as organ involvement, and medication were collected. Three nutritional assessment tools, including NAF, NT-2013, and SGA (Section A.1, Section A.2 and Section A.3), through questionnaires and physical examinations, were performed in each patient by a single assessor to minimize interpersonal variability. SGA included items related to dietary intake, weight changes, gastrointestinal symptoms, functional status, medical history of illness, and physical examination for loss of subcutaneous fat, muscle wasting, ankle edema, sacral edema, and ascites. To diagnose malnutrition, SGA relied on a combined subjective assessment of data from history and physical examination. NAF included information about current body weight, history of weight change, height, arm span, body mass index (BMI), serum albumin or total lymphocyte count if body weight was not available, dietary intake, capacity to assess food, underlying diseases, and physical examinations emphasized on general appearance. A total NAF score of 6 or more indicated malnutrition. NT-2013 incorporated details about current body weight, usual body weight, ideal body weight, change in body weight, height, patient performance status, dietary intake, underlying diseases, severity of stress affecting nutrition and metabolism, physical signs of body fat and fat loss, signs of fluid accumulation, as well as motor-power-assessed handgrip strength. NT-2013 diagnosed malnutrition when the score reached 5 or higher. The primary outcome was to compare performance in diagnosing malnutrition between NAF and NT-2013 with SGA. The secondary outcome was to determine appropriate cut-off points of NAF and NT-2013 in SSc patients.

To elucidate the efficacy and correlations of these tools, descriptive statistics, Pearson correlation analyses, and kappa coefficient of agreement were employed. Categorical variables were expressed as percentages and absolute values, while continuous variables were presented as mean ± standard deviation (SD). Pearson’s correlation coefficient (r) was used to determine the correlation between NAF and NT-2013 score [16]. Kappa (κ) statistic was calculated to measure the agreement between all assessment tools. The results were interpreted as follows: ≤0.20, poor agreement; 0.21–0.40, fair agreement; 0.41–0.60, moderate agreement; 0.61–0.80, substantial agreement; and 0.81–1.00, almost perfect agreement [17]. To analyze the sensitivity and specificity of NAF and NT-2013 for detecting malnutrition, receiver operating characteristics curves (ROC curves) were generated, including area under the curve (AUC) and their 95% confidence intervals (CI). Statistical significance was set at *p* < 0.05 for all tests. Statistical analysis was performed using SPSS version 28.0 (IBM, Armonk, NY, USA). A sample size of at least 97 participants was required to provide the level of significance of 0.05.

The study protocol was approved by the Institutional Review Board of Khon Kaen University. All participants provided written informed consent prior to enrollment in the study. Confidentiality of participant information was strictly maintained throughout the study period, and the data were anonymized for analysis.

## 3. Results

The study incorporated a total of 208 patients diagnosed with systemic sclerosis. Among these participants, 147 were identified as female, representing approximately 70.7% of the total sample, while 61 were male, accounting for approximately 29.3%. Basic characteristics were shown in Table 1.

From history taking, it was noted that a subset of patients experienced challenges in accessing food, with six patients (2.9%) reporting limitations in their ability to independently access food. Among these, two patients exhibited slight limitations, three were partially dependent, and one was entirely reliant on others for food intake. Additionally, 20 individuals (9.6%) reported significant weight loss within the past 6 months.

Gastrointestinal symptoms were prevalent among the study population, with 50 patients (24.0%) reporting symptoms upon intake. Specifically, 12 patients experienced aspiration, 42 reported dysphagia, 5 reported diarrhea, 4 reported anorexia, and 8 reported symptoms of nausea or vomiting. Importantly, it was observed that some patients experienced multiple gastrointestinal symptoms simultaneously, underscoring the complexity and multifaceted nature of gastrointestinal involvement in systemic sclerosis.

Physical examination further elucidated aspects of nutritional status and overall health. Hyposthenic build was observed in a considerable proportion of patients, with 63 individuals (30.3%) exhibiting thin or cachexic physique. Edema, indicative of fluid retention, was noted in 12 patients (5.8%), with varying degrees of severity ranging from mild to severe. Additionally, assessments of fat and muscle mass revealed significant proportions of patients with low fat mass (42 patients, 20.2%) and low muscle mass (42 patients, 20.2%) emphasizing potential sarcopenia among SSc patients.

Table 2 indicated findings that a significant proportion of the patients, specifically 95 cases (45.7%), were reportedly categorized as malnourished based on SGA. The mean and standard deviation of NAF and NT scores were 8.4 ± 4.19 and 4.0 ± 2.46, respectively. Malnutrition diagnoses using NAF and NT-2013 criteria were identified in 167 patients (80.3%) and 72 patients (34.6%), respectively.

Figure 1 showed a strong correlation between the NAF and NT-2013 total scores (r = 0.71, *p* < 0.001). However, both NAF and NT-2013 exhibited slight agreement with SGA, displaying kappa values of 0.149 for NAF and 0.131 for NT-2013. Moreover, the sensitivity and specificity of NAF for diagnosing malnutrition were determined to be 93.7% and 31.9%, respectively, while the sensitivity and specificity of NT-2013 were found to be 60.0% and 90.3%. ROC curves of NAF and NT-2013 are shown in Figure 2.

Adjusting the cut-off points of NAF and NT-2013 could enhance sensitivity, specificity, and improve agreement for diagnosis with SGA. From our study, increasing the cut-off points of NAF from 6 to 7 led to improved specificity of 69.9% (from 31.9%) while sensitivity of 89.5% remained. On the other hand, decreasing the cut-off point of NT-2013 from 5 to 4 resulted in improvement of sensitivity (60.0% from 48.4%) in exchange for less specificity (72.6% from 90.3).

## 4. Discussion

Malnutrition is common in SSc patients. In this study, the findings revealed a high prevalence of malnutrition among SSc patients, with almost half of the participants (45.7%) diagnosed as malnourished based on SGA criteria. A previous study with 56 SSc patients showed that prevalence of malnutrition, assessed by the same method, was approximately 23.2% [10]. The difference in prevalence may be due to the number of enrolled patients. Our study included 208 SSc patients, which was one of the largest sample sizes in this area of research. Our study also found that SSc patients with malnutrition had higher modified Rodnan skin scores than the well-nourished group. Dysphagia and dyspepsia tended to be more common in the malnourished group but did not reach significance. However, other abnormal gastrointestinal symptoms, duration of SSc, and other complications were not different among the two groups.

This cohort also found that prevalence of malnutrition was also varied when assessed by different nutritional assessment methods. With NAF, 80.6% of SSc patients were malnourished, while only 34.6% were reported by NT-2013. Despite the difference in prevalence, a strong correlation between NAF and NT-2013 total scores indicated that these tools assess similar aspects of nutritional status among SSc patients. Both tools included data of weight loss, abnormal GI symptoms, food intake, functional capacity, illness, physical exam, as well as illness; however, both methods emphasized different factors in calculation.

Despite the correlation of scores, both NAF and NT-2013 showed only slight agreement with SGA in diagnosing malnutrition which could be explained by several factors.

Firstly, NAF and NT-2013 emphasized different criteria compared to SGA. SGA relied heavily on subjective evaluation by healthcare providers, including physical examination and extensive patient history involving a comprehensive assessment that considered various factors beyond objective measurements, while NAF and NT-2013 focused more on objective measures categorizing nutritional status [13,14,15]. As a result, SGA provides a more holistic understanding of the patient’s nutritional status and may be more sensitive to subtle changes in nutritional status or may capture aspects of malnutrition that were not adequately addressed by NAF and NT-2013 [14].

Secondly, SGA, NAF, and NT-2013 have been developed based on populations with diverse illnesses but not SSc. SSc patients exhibited variability in their nutritional status. Factors such as disease severity, comorbidities, and individual dietary habits may contribute to discrepancies in the diagnosis of malnutrition and could impact the agreement between different assessment tools [6,7,8,9]. As a result, they may not capture certain nuances or characteristics of malnutrition that are relevant to SSc patients, leading to differences in diagnostic agreement [13,14,15].

Thirdly, SGA, NAF, and NT-2013 differed in details and the weighting of scores. Nonetheless, they shared the same underlying principle as SGA, as evidenced by the improvement in sensitivity when adjusting the cut-off points. This was due to the fact that these tools operate based on the same principle [13,14,15].

Adjusting the cut-off values of NAF and NT-2013 may improve performance of both tools in diagnosing malnutrition in SSc patients. The current cut-off point of NAF may result in overly sensitive diagnosis; increasing the threshold could improve specificity without losing sensitivity. Nonetheless, adjusting the cut-off values of NT-2013 was more challenging since improved sensitivity was accompanied by decreased specificity.

The strengths of the study included minimizing selection bias, as all eligible patients were included. Moreover, there was absence of interpersonal variability since all patients were evaluated by a single examiner. However, a single examiner could introduce potential bias in giving diagnosis due to recent experiences. Our study minimized this type of bias since NAF and NT-2013 used a scoring system; therefore, a single examiner completed only the NAF and NT-2013 checklists, but the scoring and final diagnosis were made later. Additionally, the robust sample size, considered one of the highest in this area of research, underscored the reliability of the findings. Although this study proposed a new cut-off point, the diagnostic value would be greater if there were long-term follow-ups to assess clinical outcomes.

Comorbidities of SSc may also confound the malnutrition diagnosis particularly when laboratory investigations are included. Nevertheless, NT-2013 mainly emphasized information from nutritional impact symptoms and nutritional physical exam; therefore, NT-2013 was rarely interfered with by inflammation. On the other hand, serum albumin or total lymphocyte count may be considered in NAF when data regarding body weight is not available; these markers are influenced by inflammatory processes. Fortunately, in our cohort, information about body weight was available in all participants; therefore, laboratory investigations were not considered, resulting in decreased confounding factors in our study.

Overall, while NAF and NT-2013 may correlate well with each other in assessing nutritional status among SSc patients, their agreement with SGA in diagnosing malnutrition may be influenced by various factors related to assessment approach, tool specificity, and patient variability. Adjusting the cut-off points of NAF and NT-2013 could enhance sensitivity, specificity, and improve agreement for diagnosis with SGA. Further research is still needed to understand the underlying reasons for these discrepancies and to refine nutritional assessment tools for SSc patients. Future studies could explore additional factors that may influence the diagnosis of malnutrition in SSc patients and evaluate the effectiveness of interventions aimed at improving nutritional status in this population. Long-term studies with clinical outcomes, e.g., readmission rate and mortality rate, may be needed to validate the usefulness of nutritional assessment tools. Moreover, SSc-specific nutritional assessment tools which include SSc-specific factors may also be required.

## 5. Conclusions

Malnutrition is common in SSc patients. Early detection of such a condition may lead to proper management, resulting in improved clinical outcomes. Prevalence of malnutrition in SSc patients may be varied among different nutritional assessment tools. NAF and NT-2013 exhibited a strong correlation of diagnosis of malnutrition between the two tools but only displayed slight agreement of diagnosis of malnutrition with SGA. Adjusting the cut-off points of NAF and NT-2013 could enhance sensitivity, specificity, and improve agreement for diagnosis with SGA. Further research is needed to understand the underlying reasons for these discrepancies and to refine nutritional assessment tools for systemic sclerosis patients.

## Figures and Tables

**Figure 1 life-15-01325-f001:**
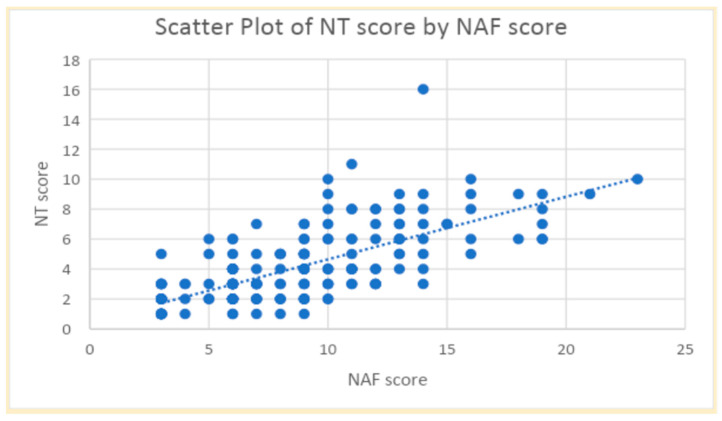
Scatter plot of NT-2013 score by NAF score.

**Figure 2 life-15-01325-f002:**
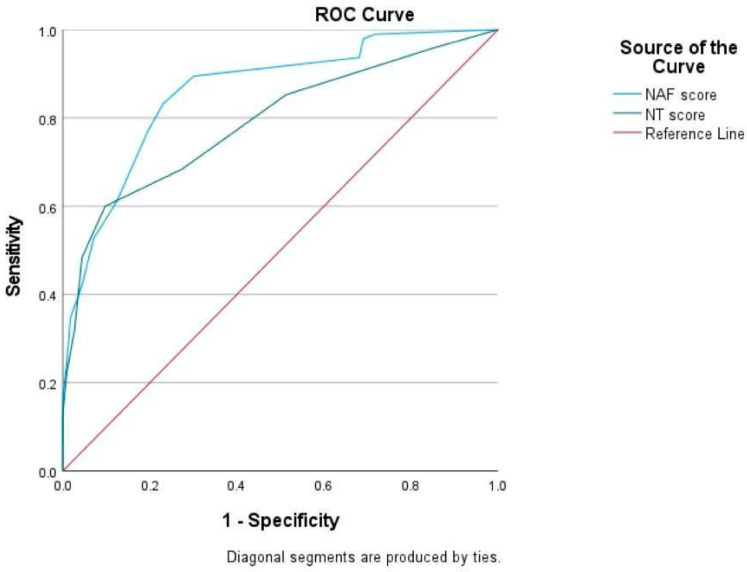
Receiver operation characteristic (ROC) curve for NAF and NT-2013.

**Table 1 life-15-01325-t001:** Characteristics of SSc patients.

Characteristics	All Patients (n = 208)	Malnutrition Classified by SGA		
		Malnourished Patients (n = 95)	Well-Nourished Patients (n = 113)	*p*-Value
Age, years (mean ± SD)	59.3 ± 11.03	61.6 ± 9.59	57.3 ± 11.80	0.005
Height, cm (mean ± SD)	157.7 ± 7.66	157.2 ± 8.41	158.0 ± 6.98	0.48
Weight, kg (mean ± SD)	52.6 ± 10.7	46.2 ± 8.90	58.0 ± 9.07	<0.001
Body mass index, kg/m^2^ (mean ± SD)	21.1 ± 3.90	18.7 ± 3.43	23.2 ± 2.98	<0.001
Duration of SSc, weeks (mean ± SD)	9.5 ± 7.57	9.2 ± 8.11	9.8 ± 7.12	0.59
Modified Rodnan skin score (mean ± SD)	6.0 ± 7.40	7.4 ± 8.71	4.8 ± 5.87	0.01
Gender (%)				0.12
-male	29.3	34.7	24.8	
-female	70.7	65.3	75.2	
Subtype of SSc (%)				0.06
-limited SSc	34.6	26.3	41.6	
-diffuse SSc	65.4	73.7	58.4	
Abnormal GI symptoms (%)				
-dry mouth	14.6	14.9	14.4	0.92
-dysphagia	20.2	25.5	15.3	0.07
-heartburn	34.6	40.4	29.7	0.11
-dyspepsia	15.1	20.2	10.8	0.06
-constipation	11.7	10.6	12.6	0.66
Serology (%)				
-positive anti Scl-70	84.5	84.6	84.4	0.97
-positive anticentromere	12.3	13.2	11.6	0.76
Complications (%)				
-overlapping syndrome	9.13	9.5	8.8	0.98
-myositis	6.3	8.4	4.4	0.49
-pulmonary fibrosis	45.0	44.2	46.0	0.96
-pulmonary hypertension	6.3	5.3	7.4	0.82
-renal crisis	0.48	1.1	0	0.55

GI: gastrointestinal tract; SGA: subjective global assessment; SSc: systemic sclerosis.

**Table 2 life-15-01325-t002:** Malnutrition diagnosed by SGA, NAF, and NT-2013.

Nutritional Assessment Tool	Mean Score (SD)	Current Cut-Off Point for Malnutrition	Prevalence of Malnutrition by Using Current Cut-Off (N = 208)
Subjective Global Assessment (SGA)	NA	No score (using global assessment)	45.7%
Nutrition Alert Form (NAF)	8.4 (4.19)	6 or more	80.3%
Nutrition Triage 2013 (NT-2013)	4.0 (2.46)	5 or more	34.6%

## Data Availability

The data presented in this study are available on request from the corresponding author due to privacy and ethical reasons.

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
