# Peer review of "Comparing Performance of NAF and NT-2013 to SGA as Nutritional Assessment Tools in Systemic Sclerosis Patients"

_life, 2025, doi:10.3390/life15081325_

Round 1

Reviewer 1 Report

Comments and Suggestions for Authors
  • Line 51; it is not clear what the authors wanted to mean by ‘insults’.
  • Line 76; ‘ACR/EULAR criteria’, which I assume the version in 2013, should be precisely described so even non-rheumatologists can grasp the meaning.
  • The content of display items needs much improvement. Patient group with SSc consist of a heterogeneous population, and disease phenotype is much altered disease-specifific autoantibodies including anti-Topoâ… , centromere and RNA polymerase â…¢ The authors should show disease duration, results of modified total skin score and autoantibody profile in the study patients, and should perform analyses between these disease activity markers/determinants and NAF/NT-2013. How many patients were diagnosed as having gastrointestinal involvement of SSc by endoscopy?
  • No control groups?

Author Response

Dear editors and reviewers,

First of all, we would like to express our gratitude for the valuable comments, which we have tried our best to improve our manuscript. 

Comment 1) Line 51; it is not clear what the authors wanted to mean by “insults”.

Response: Thank you for point this out. We agree and have revised the sentence. “The parameters could be interfered by various illness-related factors leading to a large number of misdiagnosis of malnutrition.” (P. 2 Line 51)

Comment 2) Line 76; “ACR/EULAR criteria”, which I assume the version 2013, should be precisely described so even non-rheumatologists can grasp the meaning.

Response: Thank you for your comment. We agree and have added the following to the manuscript. ”Regarding ACR/EULAR 2013 criteria, skin thickening of the fingers extending proximal to the metacarpophalangeal joints is sufficient to be classified as SSC, if not, seven additive items are considered with different weights for each: skin thickening of the fingers, finger-tip lesions, telangiectasia, abnormal nailfold capillaries, interstitial lung disease or pulmonary arterial hypertension, Raynaud’s phenomenon, and SSc-related autoantibodies.” (P. 2-3 Line 78-83)

Comment 3) The content of display items needs much improvement. Patient group with SSc consist of a heterogenous population, and disease phenotype is much altered disease-specific autoantibodies including anti-TopoI, centromere, and RNA polymerase III. The authors should show disease duration, results of modified total skin score and autoantibody profile in the study patients, and perform analyses between these disease activity markers/determinants and NAF/NT-2013.

Response: Thank you for your suggestion. We agree and have revised table 1 (P. 4 Line 130) and have added some elaboration. “Our study also found that SSc patients with malnutrition had higher modified Rodnan skin score than well-nourished group. Dysphagia and dyspepsia tended to be more common in malnourished group, but did not reach the significance.  However, other abnormal gastrointestinal symptoms, duration of SSc and other complications were not different among 2 groups.” (P. 7 Lin 184-188)           

            In the table, we also perform analyses of SSc activity markers and manifestations and SGA in addition to NAF and NT-2013 because, currently, SGA is an international standard nutritional assessment tool, while NAF and NT-2013 are new local methods which required further validation.

Regarding SSc-related autoantibodies, rate of positive anti Scl-70 is rather high in Thailand. Although this was not a nationwide study, previous cohort studies from other SSc centers in Thailand have reported similar results as the following references.

Khon Kaen University (Northeastern Thailand)

  1. Foocharoen C, Peansukwech U, Mahakkanukrauh A, Suwannaroj S, Pongkulkiat P, Khamphiw P, Nanagara R. Clinical characteristics and outcomes of 566 Thais with systemic sclerosis: A cohort study. Int J Rheum Dis. 2020 Jul;23(7):945-957.
  1. Foocharoen C, Watcharenwong P, Netwijitpan S, Mahakkanukrauh A, Suwannaroj S, Nanagara R. Relevance of clinical and autoantibody profiles in systemic sclerosis among Thais. Int J Rheum Dis. 2017 Oct;20(10):1572-1581.

Chiang Mai University (Northern Thailand)

  1. Wangkaew S, Prasertwittayakij N, Euathrongchit J. Clinical Manifestation and Incidence of Cardiopulmonary Complications in Early Systemic Sclerosis Patients with Different Antibody Profiles. J Clin Med Res. 2019 Jul;11(7):524-531.
  1. Wangkaew S, Euathrongchit J, Wattanawittawas P, Kasitanon N, Louthrenoo W. Incidence and predictors of interstitial lung disease (ILD) in Thai patients with early systemic sclerosis: Inception cohort study. Mod Rheumatol. 2016 Jul;26(4):588-93.

Mahidol University (Central Thailand)

  1. Guayboon T, Muangchan C. Prevalence of and factors independently associated with digital ischemic complications in patients with systemic sclerosis. J Scleroderma Relat Disord. 2023 Feb;8(1):43-52.
  1. Punjasamanvong S, Muangchan C. Persistent eosinophilia and associated organ involvement in Thai patients with systemic sclerosis: Data from the Siriraj scleroderma cohort. Arch Rheumatol. 2021 Oct 18;36(4):527-537.
  1. Likhit O, Louthrenoo W, Pattanakitsakul SN, Suttitheptumrong A, Hannongbua S, Rungrotmongkol T, Noguchi H, Takeuchi F, Boonnak K. Determination of T Cell Responses in Thai Systemic Sclerosis Patients. J Immunol Res. 2022 Mar 7;2022:5072154.

Thammasat University (Central Thailand)

Pirompanich P, Sathitakorn O, Sakulvorakitti T. Pulmonary function in Thai patients with systemic sclerosis; a single center 6-year retrospective study. F1000Res. 2024 May 31;13:296.

            Due to the low prevalence and limited clinical relevance of other specific autoantibody profiles (such as anti-RNA polymerase III, anti-Th/To, anti-U3RNP) in Thai patients with SSc, these antibodies were not included as covariates in the statistical analysis. [Foocharoen C, Watcharenwong P, Netwijitpan S, Mahakkanukrauh A, Suwannaroj S, Nanagara R. Relevance of clinical and autoantibody profiles in systemic sclerosis among Thais. Int J Rheum Dis. 2017 Oct;20(10):1572-1581.]

Comment 4) How many patients were diagnosed as having gastrointestinal involvement of SSc by endoscopy?

Response: Since invasive procedures may be limited in some patients, esophageal involvement is defined when any esophageal symptoms of SSc presented (i.e., esophageal dysphagia, heartburn, or reflux symptoms). [Foocharoen C, Peansukwech U, Mahakkanukrauh A, Suwannaroj S, Pongkulkiat P, Khamphiw P, Nanagara R. Clinical characteristics and outcomes of 566 Thais with systemic sclerosis: A cohort study. Int J Rheum Dis. 2020 Jul;23(7):945-957.]

In this study, 20% of SSc patients had dysphagia, 34.6% with heartburn, and 15.1% with dyspepsia (table 1).

Comment 5) No controls groups?

Response: Since this study was conducted in the SSc clinic, all participants had SSc. The main objective was to determine the utility of nutritional assessment tools, therefore, the analysis compared baseline characteristics between malnourished and well-nourished group was performed (table 1).

Best regards,

Veeradej Pisprasert

Reviewer 2 Report

Comments and Suggestions for Authors

Summary of the article

The introduction highlights systemic sclerosis (SSc) as a rare connective tissue disease with significant morbidity due to complications like gastrointestinal involvement, which often leads to malnutrition. Malnutrition in SSc patients is linked to poor quality of life and increased mortality, necessitating reliable assessment tools. While Subjective Global Assessment (SGA) is the gold standard, its subjectivity and time-consuming nature have prompted the exploration of alternatives like Thailand’s Nutritional Alert Form (NAF) and Nutritional Triage 2013 (NT-2013). The study aims to compare these tools’ performance against SGA in diagnosing malnutrition among SSc patients.

A cross-sectional study enrolled 208 adult SSc patients from a Thai hospital. Nutritional status was assessed using SGA, NAF, and NT-2013 by a single examiner to minimize variability. Data included demographics, clinical symptoms, and physical examinations. Statistical analyses (Pearson correlation, Kappa coefficient, ROC curves) evaluated tool agreement, sensitivity, and specificity. The study also explored adjusting cut-off points for NAF and NT-2013 to optimize diagnostic accuracy.

Malnutrition prevalence varied by tool: 45.7% (SGA), 80.3% (NAF), and 34.6% (NT-2013). NAF showed high sensitivity (93.7%) but low specificity (31.9%), while NT-2013 had moderate sensitivity (60.0%) and high specificity (90.3%). Both tools correlated strongly (r=0.71) but agreed poorly with SGA (kappa: 0.149 for NAF, 0.131 for NT-2013). Adjusting cut-offs improved performance—raising NAF’s threshold increased specificity (69.9%), and lowering NT-2013’s threshold improved sensitivity (48.4% to 60.0%).

Areas for improvement

  • The study notes that NAF and NT-2013 were not designed for SSc. Future research could adapt these tools to SSc-specific factors (e.g., dysphagia severity, malabsorption) to improve relevance.
  • The cross-sectional design limits insights into how malnutrition diagnoses correlate with clinical outcomes (e.g., disease progression, mortality). A longitudinal follow-up would strengthen the findings.
  • The study was conducted at a single center in Thailand. Including multi-center or international cohorts could enhance generalizability.
  • While adjusting cut-offs improved metrics, the trade-offs (e.g., sensitivity vs. specificity) warrant deeper analysis. A larger validation cohort could refine optimal thresholds.
  • Comparing NAF/NT-2013 with other global tools (e.g., GLIM criteria) could provide broader context for their utility in SSc.
  • Subgroup analyses by SSc subtypes (diffuse/limited) or disease severity might reveal tool performance variations.
  • Incorporating patient-reported outcomes (e.g., symptom burden) could align objective measures with subjective experiences of malnutrition.
  • The discussion should explicitly address potential biases (e.g., single-examiner bias) and confounding factors (e.g., comorbidities affecting nutrition).

The study provides valuable insights into nutritional assessment challenges in SSc but underscores the need for SSc-specific tool adaptations and outcome-linked validation. Addressing these gaps could enhance clinical utility and patient care.

Author Response

Dear editors and reviewers,

First of all, we would like to express our gratitude for the valuable comments, which we have tried our best to improve our manuscript with the following explanations.

Comment 1) This study noted that NAF and NT-2113 were not designed for SSc. Future research could adapt these tools to SSc-specific factors (e.g. dysphagia severity, malabsorption) to improve relevance. The cross-sectional design limits insights into how malnutrition diagnoses correlate with clinical outcomes (e.g. disease progression, mortality). A longitudinal follow-up would strengthen the findings.

Response: Thank you for your suggestion. We agreed and have added in the discussion to emphasize this point. “Long-term study with clinical outcomes, e.g. readmission rate, mortality rate, may be needed to validate the usefulness of nutritional assessment tools. Moreover, SSc-specific nutritional assessment tools which include SSc-specific factors may also be required.” (P. 8 Line 250-253)

Comment 2) This study was conducted in a single center in Thailand. Including multi-center or international cohorts could enhance generalizability.

Response: Thank you for your comment. Although this was not a nationwide study, previous cohort studies from other SSc centers in Thailand have reported similar results in several aspects as the following references. However, we will consider your suggestion to include muti-center study in the next steps.

Khon Kaen University (Northeastern Thailand)

  1. Foocharoen C, Peansukwech U, Mahakkanukrauh A, Suwannaroj S, Pongkulkiat P, Khamphiw P, Nanagara R. Clinical characteristics and outcomes of 566 Thais with systemic sclerosis: A cohort study. Int J Rheum Dis. 2020 Jul;23(7):945-957.
  1. Foocharoen C, Watcharenwong P, Netwijitpan S, Mahakkanukrauh A, Suwannaroj S, Nanagara R. Relevance of clinical and autoantibody profiles in systemic sclerosis among Thais. Int J Rheum Dis. 2017 Oct;20(10):1572-1581.

Chiang Mai University (Northern Thailand)

  1. Wangkaew S, Prasertwittayakij N, Euathrongchit J. Clinical Manifestation and Incidence of Cardiopulmonary Complications in Early Systemic Sclerosis Patients with Different Antibody Profiles. J Clin Med Res. 2019 Jul;11(7):524-531.
  1. Wangkaew S, Euathrongchit J, Wattanawittawas P, Kasitanon N, Louthrenoo W. Incidence and predictors of interstitial lung disease (ILD) in Thai patients with early systemic sclerosis: Inception cohort study. Mod Rheumatol. 2016 Jul;26(4):588-93.

Mahidol University (Central Thailand)

  1. Guayboon T, Muangchan C. Prevalence of and factors independently associated with digital ischemic complications in patients with systemic sclerosis. J Scleroderma Relat Disord. 2023 Feb;8(1):43-52.
  1. Punjasamanvong S, Muangchan C. Persistent eosinophilia and associated organ involvement in Thai patients with systemic sclerosis: Data from the Siriraj scleroderma cohort. Arch Rheumatol. 2021 Oct 18;36(4):527-537.
  1. Likhit O, Louthrenoo W, Pattanakitsakul SN, Suttitheptumrong A, Hannongbua S, Rungrotmongkol T, Noguchi H, Takeuchi F, Boonnak K. Determination of T Cell Responses in Thai Systemic Sclerosis Patients. J Immunol Res. 2022 Mar 7;2022:5072154.

Thammasat University (Central Thailand)

Pirompanich P, Sathitakorn O, Sakulvorakitti T. Pulmonary function in Thai patients with systemic sclerosis; a single center 6-year retrospective study. F1000Res. 2024 May 31;13:296.

Comment 3) While adjusting cut-offs improved metrics, the trade-offs (e.g., sensitivity vs. specificity) warrant deeper analysis. A larger validation cohort could refine optimal thresholds.

Response: Thank you for pointing this out. We agree and have planned to conduct larger cohort and/or multi-center study in the near future.

Comment 4) Comparing NAF/NT-2013 with other global tools (e.g., GLIM criteria) could provide broader context for their utility in SSc.

Response: Thank you for your suggestion. We agree that studying with global tools such as GLIM criteria is importance. However, objective muscle mass measurement was not performed in this study. Several patients had a problem of stiff joints resulting in inability to perform bioelectrical analysis (BIA) measurement with our instrument in our center which required holding a handheld device. However, we plan to study usefulness of GLIM-criteria in the future when another model of BIA machine may be more feasible.

Comment 5) Subgroup analyses by SSc subtypes (diffuse/limited) or disease severity might reveal tool performance variations.

Response: Thank you for your comment. We agree and have added information about SSc subtypes along with SSc manifestations in table 1 (P. 4 Line 130). Comparison of these factors between malnourished and well-nourished was also performed.

Comment 6) Incorporating patient-reported outcomes (e.g., symptom burden) could align objective measures with subjective experiences of malnutrition.

Response: Thank you for pointing this out. We agree and have added information about SSc manifestations, particularly abnormal GI symptoms, in table 1 (P. 4 Line 130).

Comment 7) The discussion should explicitly address potential biases (e.g., single-examiner bias) and confounding factors (e.g., comorbidities affecting nutrition).

Response: Thank you for your comment. We agree and have added further discussion. “However, a single examiner could introduce potential bias in giving diagnosis due to recent experiences. Our study minimized this type of bias since NAF and NT-2013 used a scoring system, therefore, a single examiner completed only the NAF and NT-2013 checklists, but the scoring and final diagnosis were made later.” (P. 8 Line 225-228) “Comorbidities of SSc may also confound the malnutrition diagnosis particularly when laboratory investigations were included. Nevertheless, NT-2013 mainly emphasized on information from nutritional impact symptoms and nutritional physical exam, therefore, NT-2013 was rarely interfered by inflammation. On the other hand, serum albumin or total lymphocyte count may be considered in NAF when data regarding body weight was not available, these markers were influenced by inflammatory process. Fortunately, in our cohort, information about body weight was available in all participants, therefore, laboratory investigations were not considered, resulting in decreased confounding factors in our study.” (P. 8 Line 233-241)

Comment 8) The study provides valuable insights into nutritional assessment challenges in SSc but underscores the need for SSc-specific tool adaptations and outcome-linked validation. Addressing these gaps could enhance clinical utility and patient care.

Response: Thank you for your suggestion. We agreed and have added in the discussion. “Long-term study with clinical outcomes, e.g. readmission rate, mortality rate, may be needed to validate the usefulness of nutritional assessment tools. Moreover, SSc-specific nutritional assessment tools which include SSc-specific factors may also be required.” (P. 8 Line 250-253)

Best regards,

Veeradej Pisprasert

Round 2

Reviewer 1 Report

Comments and Suggestions for Authors

(No further comments)

Reviewer 2 Report

Comments and Suggestions for Authors

Introduction: Systemic sclerosis (SSc) is a rare connective tissue disease characterized by vasculopathy and fibrosis, often leading to gastrointestinal complications and malnutrition, which significantly impact morbidity and quality of life. While Subjective Global Assessment (SGA) is the gold standard for malnutrition diagnosis, its subjectivity and time-consuming nature have prompted the use of simpler tools like Thailand’s Nutritional Assessment Form (NAF) and Nutritional Triage 2013 (NT-2013). However, these tools were not designed for SSc, raising questions about their validity in this population. This study aimed to compare NAF and NT-2013 against SGA in diagnosing malnutrition among SSc patients.

Methods: A cross-sectional study enrolled 208 adult SSc patients from a Thai tertiary hospital. Nutritional status was assessed using SGA, NAF, and NT-2013, with data collected on demographics, disease characteristics, and gastrointestinal symptoms. Agreement between tools was analyzed using kappa statistics, while sensitivity, specificity, and ROC curves evaluated diagnostic performance. Cut-off adjustments for NAF and NT-2013 were explored to optimize agreement with SGA.

Results: Nearly half (45.7%) of patients were malnourished per SGA, while NAF and NT-2013 identified malnutrition in 80.3% and 34.6%, respectively. NAF showed high sensitivity (93.7%) but low specificity (31.9%), whereas NT-2013 had moderate sensitivity (60.0%) and high specificity (90.3%). Both tools correlated strongly (r=0.71) but agreed poorly with SGA (kappa: 0.149 for NAF, 0.131 for NT-2013). Adjusting NAF’s cut-off to ≥7 improved specificity (69.9%), while lowering NT-2013’s threshold to ≥4 increased sensitivity (60.0%). Malnourished patients had lower BMI, higher skin scores, and more frequent dysphagia/dyspepsia.

Review of Revisions and Responses: 

The authors responded constructively to reviewer feedback, addressing key concerns while acknowledging limitations. They appropriately emphasized future directions, such as developing SSc-specific tools and longitudinal validation, but could have proposed concrete adaptations (e.g., weighting dysphagia severity in scoring). Their defense of single-center data by citing similar Thai cohorts was reasonable, though generalizability remains limited without multi-center validation. They acknowledged cut-off trade-offs and GLIM comparison barriers. The discussion of bias and confounders was thorough, but comorbidities’ role in malnutrition was under-explored. Overall, the revisions improved the manuscript’s rigor.